# Aggregative Swab Sampling Method for Romaine Lettuce Show Similar Quality and Safety Indicators and Microbial Profiles Compared to Composite Produce Leaf Samples in a Pilot Study

**DOI:** 10.3390/foods13193080

**Published:** 2024-09-27

**Authors:** Rachel J. Gathman, Jorge Quintanilla Portillo, Gustavo A. Reyes, Genevieve Sullivan, Matthew J. Stasiewicz

**Affiliations:** 1Department of Food Science and Human Nutrition, University of Illinois Urbana-Champaign, Urbana, IL 61801, USA; 2SmartWash Solutions, LLC, Salinas, CA 93908, USA

**Keywords:** preharvest, leafy greens, microbial community, sampling

## Abstract

Composite produce leaf samples from commercial production rarely test positive for pathogens, potentially due to low pathogen prevalence or the relatively small number of plants sampled. Aggregative sampling may offer a more representative alternative. This pilot study investigated whether aggregative swab samples performed similarly to produce leaf samples in their ability to recover quality indicators (APCs and coliforms), detect *Escherichia coli*, and recover representative microbial profiles. Aggregative swabs of the outer leaves of romaine plants (*n* = 12) and composite samples consisting of various grabs of produce leaves (*n* = 14) were collected from 60 by 28 ft sections of a one-acre commercial romaine lettuce field. Aerobic plate counts were 9.17 ± 0.43 and 9.21 ± 0.42 log(CFU/g) for produce leaf samples and swabs, respectively. Means and variance were not significantly different (*p* = 0.38 and *p* = 0.92, respectively). Coliform recoveries were 3.80 ± 0.76 and 4.19 ± 1.15 log(CFU/g) for produce leaf and swabs, respectively. Means and variances were not significantly different (*p* = 0.30 and *p* = 0.16, respectively). Swabs detected generic *E. coli* in 8 of 12 samples, more often than produce leaf samples (3 of 14 positive, Fisher’s *p* = 0.045). Full-length 16S rRNA microbial profiling revealed that swab and produce leaf samples did not show significantly different alpha diversities (*p* = 0.75) and had many of the most prevalent bacterial taxa in common and in similar abundances. These data suggest that aggregative swabs perform similarly to, if not better than, produce leaf samples in recovering indicators of quality (aerobic and coliform bacteria) and food safety (*E. coli*), justifying further method development and validation.

## 1. Introduction

Each year in the United States, it is estimated that 31 major foodborne pathogens cause 9.4 million cases of foodborne illnesses [1]. Nearly half of foodborne illnesses from 1998 to 2008 were attributed to produce, and leafy greens alone were associated with 22% of illnesses, more than any other commodity [2]. Additionally, a study examining foodborne outbreaks and illnesses in the United States and Puerto Rico from 2009–2015 attributed 77 outbreaks to leafy greens, which represented 6% of the food reported outbreaks attributed to a single food category [3]. Furthermore, one model attributed 228,000 illnesses per year to romaine lettuce with an estimated economic burden of USD 728 million per year [4]. One pathogen that causes outbreaks linked to leafy greens is Shiga-toxin-producing *Escherichia coli* (STEC). STEC can contaminate leafy greens through contaminated soil [5]. Contamination can also occur via cattle and other animals directly through shedding or indirectly by contaminated runoff, irrigation water, and dust [6,7,8,9]. The Centers for Disease Control and Prevention National Outbreak Reporting System reported 18 foodborne outbreaks, 854 illnesses, 323 hospitalizations, and 5 deaths from 2009–2020 due to *Escherichia* associated with romaine lettuce [10]. Additionally, one study found that of the leafy green-associated STEC outbreaks from 2009 to 2018, romaine lettuce was identified as the source of the outbreaks more often than other types of leafy greens [11]. Of the 40 confirmed or suspected outbreaks from STEC in leafy greens that were investigated, 80% were caused by STEC O157, 8% O145, 5% O26, 3% O111, 3% O126, and 1 caused by both O157 and O26 [11]. Romaine lettuce may especially be at risk for contamination possibly due to characteristics such as its shape and physiology [11]. For example, romaine has loosely clumped leaves and an open top that may make the plant more vulnerable to STEC contamination compared to other leafy greens, such as iceberg lettuce, which has a tighter formation of leaves [11]. While STEC and other pathogens cause outbreaks in other commodities such as meat and dairy, leafy greens can present additional challenges since they are typically consumed raw with no cooking or thermal process being conducted before consumption to eliminate microorganisms [5,7]. Additionally, they are typically grown near the soil’s surface which can place the produce at a higher risk of cross-contamination from the soil [5,7]. Furthermore, produce can be difficult to sanitize once contaminated [5]. Therefore, preventing contamination of leafy greens during preharvest is critical for reducing associated illnesses and outbreaks [7]. As an indicator of the level of concern regarding produce safety, the FDA issued the “Produce Safety Rule” in 2015 as a rule with its authority in the Food Safety Moderation Act that is designed to enhance preharvest preventative practices by adding requirements for personnel qualification and training, health and hygiene, agricultural water, biological soil amendments, domesticated and wild animals, and equipment, tools, and buildings [12]. 

Preharvest sampling of romaine lettuce for food safety testing often consists of collecting and compositing 60 grab samples of romaine leaves, ranging from 150 to 375 g total mass, using either a random or predetermined sampling locations [13]. However, within the field, pathogens can be present in low concentrations or be dispersed heterogeneously among a field which can make their detection more challenging and makes selecting sampling methods that increase the statistical power of food safety testing critical [14,15,16]. To obtain reliable detection of contaminated produce, even larger numbers of composite samples would need to be collected preharvest than is typical [16]. For example, many plans specify compositing 60 grabs samples usually of ≤375 g total mass, referred to casually as an ‘N60’ plan. This is an adaptation of the most stringent ICMSF recommendations for sampling plans (Case 15), which specify testing 60 individual 25-gram samples, and rejecting the lot if any test positive. If 1% of the samples were contaminated, then probability calculations show a plan with these 60 samples would have only a 45% chance of detecting the contamination. Further, the 1% positivity is casually referred to as ‘prevalence’ to combine all the confounding factors leading to a positive sample, including sample mass and assay limits of detection. All else being equal, taking a larger number of samples would decrease the probability of acceptance [17]. For example, 298 samples would give a 95% chance of detecting contamination at 1% prevalence [18]. Additionally, one study found that the theoretical detection probability in fields with low prevalence of contamination (<3.7%) decreases below 20% when collecting ≤five samples [14]. These probabilities were calculated based on the number of samples collected and the number of contamination sites in the field [14]. Furthermore, the study also found that the detection probability decreases in fields with smaller contaminated areas or when the sampled area becomes smaller [14]. Therefore, a sampling method that covers a larger sampling area and is more representative of the field might increase the chance of detecting contamination at varying levels of prevalence. 

In the beef industry, a more representative and nondestructive sampling method compared to N60 excision sampling consisting of collecting aggregative swabs has been tested and approved for use in detecting naturally occurring *E. coli* O157:H7 and *Salmonella* in beef trims [19,20]. This method involves using a swab made of spun bound olefin polymer cloth material to sample the beef trims. The swabs had similar organism recovery and detection of pathogens, their surrogates, or pathogen index targets, and were at least as accurate as N60 excision sampling (i.e., 60 surface excision slices with a total sample mass of approximately 375 g per lot) [19,20]. Two different approaches were used for obtaining the swabs of the beef trim: (i) continuous sampling obtained by positioning a device to secure the swab at the end of a conveyor belt to allow trim pieces to rub on the swab while falling into the bins and (ii) manually sampling the trim at the top of the bins by hand through pushing and rubbing the swab against the meat [19,20]. It is estimated that the manual method samples approximately 11,000 cm^2^ of meat, while the N60 excision method samples approximately 1100 cm^2^ [19,20]. This aggregative sampling method has received a FSIS Letter of No Objection for use of nondestructive sampling of beef trim for *Salmonella,* STEC, and indicator organisms [21]. In this study, the novel manual swab sampling technique used in the beef industry was applied for use in romaine lettuce fields. 

A major advantage of this novel sampling method is that the swab can be passed over a romaine field, creating a sample representative of hundreds to thousands of plants compared to a composite grab sample of produce leaves representative of typically just 60 plants. Additionally, the current practice of collecting composite produce leaf samples for food safety testing already requires sample collectors to walk through the field to complete a specified sampling plan, meanwhile walking past hundreds or thousands of plants to grab produce leaves. Alternatively, an aggregative swab sample could be collected using the same or a similar sampling pattern as what is already used for composite sample collection resulting in a sample that does not require any additional time to collect compared to current practices and is representative of potentially hundreds of more plants. Additionally, although the swabs could present potential for transferring contamination from plant to plant, the aggregative swab sampling method has received an FSIS Letter of No Objection for use in aggregately sampling combo bins of beef trim for food safety testing [21], which also presents the same opportunity for transferring contamination. The end goal of the aggregative swab sampling method is to detect zero-tolerance organisms, i.e., a specific organism that if found in any amount makes the product be considered adulterated and unfit for human consumption. In the case of a zero-tolerance organism being detected, the whole lot of product would be rejected. Therefore, it does not matter if the swab sample is transferring contamination from plant to plant, since the swab would then presumably test positive and result in the rejection of the entire lot.

This pilot study was part of a larger study of *E. coli* presence in soil, air, and plant leaves from fields with years of commercial production. The commercial field used in this study was managed as if it had been harvested for sale and consumption. However, it was earmarked for research purposes. The value of working in a commercial field setting is working on industry-relevant product. However, coordinating with industry collaborators with existing systems added complexity to the study and required some compromises, mainly limited control in study design, sample collection, processing methods, and information available for publication that might not have been necessary if this was done in an academic field setting. While this study has limitations, working with industry collaborators allowed for testing at scale on industry relevant product and allowed for investigatory proof-of-concept experiments to determine if future work is justified. Therefore, we report this work as pilot study to prove the concept of the aggregative swab sampling method for romaine lettuce and to justify future work and investments in equipment and labor required to test and develop this method more thoroughly for food safety applications at commercial scale. 

For this pilot study, composite romaine leaf samples were collected using stratified random and systematic patterns as well as aggregative swabs of romaine. While similar field trials were conducted by Quintanilla et al. for validation of a simulation model [13], this study’s main goal was to serve as a proof-of-concept experiment for the aggregative swab sampling method for leafy greens to justify future development and work including pathogen detection. Additionally, while a similar aggregative sampling approach has been tested for soil sampling within romaine fields [22], this pilot study is the first reported use of an aggregative swab sample being collected directly from leafy green tissue and used for food safety testing. For this study, in order to determine whether the swabs could be a potentially viable sampling method in romaine lettuce, tests were conducted to see if the swabs could recover similar, if not higher, concentrations of aerobic bacteria and coliforms, determine if the microbial community profiles of the swabs were representative of the produce leaves, and compare the capability of swabs and produce leaf samples to pick up and detect *E. coli* from uninoculated field areas and from areas spray inoculated with a rifampicin-resistant *E. coli* cocktail. This was performed as a test of the swabs’ representativeness of the produce leaves and to help gain confidence that if a pathogen were present on the produce in future studies, the swab sampling method has the potential to recover and detect them.

## 2. Materials and Methods

### 2.1. Commercial Field Set Up and Inoculation

The field was operated by a commercial grower in the Salinas Valley of California. The following methods regarding the field setup, inoculation, and sample collection were written from information that the industry collaborators were able to share.

An acre portion of the field was split into three sections with each section being further subdivided into two subsections. Each subsection measured approximately 60 ft 4 in (18.4 m) by 28 ft 3 in (8.6 m). Each of the three sections contained one uninoculated subsection and one subsection inoculated with a cocktail of rifampicin-resistant *E. coli*. The cocktail inoculum was prepared by AMTEK Inc., a laboratory in Fremont, CA, USA. The cocktail consisted of the *E. coli* strains TVS 353, TVS 354, and TVS 355. TVS 353 was originally isolated from surface irrigation water, TVS 354 from romaine lettuce, and TVS 355 from sandy-loam soil near Salinas, California [23]. The *E. coli* strains were grown using tryptic soy agar supplemented with 0.1 g/L rifampicin and were incubated at 37 °C for 18–24 h. These cultures were used for the inoculation of the plants. The plants were mature and ready for harvest, approximately 10 to 12 weeks. The concentration of the inoculum was 9 log(CFU/mL) and was diluted to the target concentration of 7 log(CFU/mL). Each of the three sections used for this study were spray-inoculated on a different day. Each inoculated subsection was inoculated by spraying 2 L of inoculum onto the plants using a garden sprayer. Section one was the first to be inoculated. Section two was inoculated three days later, and section three was inoculated two days after section two. Samples were collected on the same day at 0-, 2-, and 5-days post inoculation. 

### 2.2. Commercial Field Sample Collection

Samples collected from each uninoculated and inoculated subsection were: (i) two aggregative swabs, (ii) one composite produce leaf sample taken in a stratified random pattern and (iii) at least one composite produce leaf taken in a systematic pattern. The swabs were made of spun bound polymer cloth material measuring 61 by 21 cm which came sterile and individually packaged (MicroTally^®^, Fremonta Corporation. San Jose, CA, USA). Each swab was pre-wet by spraying with approximately 10 mL of distilled water before collection. Swabs were then attached to a floor duster (Swiffer, Procter & Gamble Company, Cincinnati, OH, USA) with duster head measuring 10 in. (25.4 cm) by 4.8 in (12.2 cm). The floor duster was pressed up against the tops of the plants and gently dragged across the romaine plants to avoid damage. This was done across the width of each subsection (approximately 28 ft 3 in, or 8.6 m). No noticeable damage to plants was observed by sample collectors or other workers immediately following collection of the swabs. For the systematic sampling, a grab of produce leaves was taken every *k* step (i.e., the spacing of the collection of every grab of leaves was distributed evenly among the field and taken after a certain number of steps). For the stratified random sampling, each of the six subsections were divided into three strata and three grabs of leaves were randomly taken from each strata. (Appendix A). For both produce leaf sample collection patterns, leaves were taken by manually grabbing the leaves off the plant with gloves hands. After collection, samples were placed in coolers with ice packs and shipped to University of Illinois Urbana-Champaign for processing.

### 2.3. Inoculation and Sampling of Store-Bought Romaine as an Outgroup for Analysis

Store-bought romaine was used as an outgroup to compare to the field samples. The outgroup was used as a contrast to help demonstrate the similarity between the swab and produce samples from the field. The comparison of the store-bought romaine samples to the swab and produce samples from the field were not used to determine the conclusions of the study or next steps. The preparation of the store-bought samples was done at a University of Illinois Urbana Champaign lab. Prepackaged heads of romaine lettuce were bought from a local chain grocery store. A total of twelve heads were bought weighing approximately 150 g each. Six heads were inoculated. The inoculum was prepared using a different media and *E. coli* strain compared to the field trial due to different labs preparing the inoculum. The inoculum consisted of *E. coli* K12 (ATCC 29425). The *E. coli* was grown in BHI broth at 37 °C with agitation for 18–24 h. The broth culture was diluted a thousandfold with PBS to obtain a target inoculum level of 6–7 log, with a measured inoculum concentration of 6.38 log(CFU/mL). The whole lettuce heads were placed individually in a WhirlPak bag, inoculated with 150 mL of inoculum, mixed, allowed to stand for 2–5 min, drained of excess, and allowed to dry in the biosafety cabinet for 2 h before sampling. We used direct dilution of the BHI culture, not a washed pellet, for simplicity, because this inoculum was only used to prepare an outgroup for sampling after 2 h of acclamation—the residual nutrients from depleted BHI are unlikely to shift the microbiome in such a short time period. 

Sampling of store-bought samples included three inoculated produce leaf samples, three swabs of inoculated heads, three uninoculated produce leaf samples, and three swabs of uninoculated heads. Swabs were pre-wet with 10 mL of distilled water. Swabs were held with fresh nitrile gloves that were sprayed with 70% ethanol and allowed to dry, and the entire outside of the romaine heads were sampled with the swab. Produce leaf samples were taken by cutting pieces of outer leaves from various outer side and top portions of the plant with sterile forceps. Swab and produce leaf samples were further processed as described below. 

### 2.4. Sample Processing

For both leaf and swabs, a starting 1:10 dilution of sampled matter was made and used for enumeration, *E. coli* presence absence, and DNA extraction. For both field and store produce leaf samples, 25 g of romaine leaves were mixed with 225 mL of PBS buffer in a WhirlPak bag and massaged by hand for an initial 1:10 dilution. For swabs, the weight of whatever was picked up by the swab during aggregative sampling (i.e., leaf tissue, dirt, etc.) was calculated by massing the sample swab and subtracting the average sterile swab mass (10.5 g ± 0.2 g). The mass of picked up matter and water used to moisten the swabs ranged from 10.8 to 18.2g, and these weights were used to determine the amount of PBS to add to create a 1:5 dilution. The 10 mL of water used to moisten the swab was considered an initial 1:2 dilution. Therefore, the 1:5 dilution with PBS based off of the mass of picked up matter created the starting 1:10 dilution (Appendix A). A portion of the PBS washes were serially diluted for plate counts, while an aliquot of the original dilution was saved for later microbial community analysis. Aliquots from commercial field samples were stored in 50 mL centrifuge tubes at 4 °C for approximately three months prior to DNA extraction due to logistical reasons. Specifically, the need to organize supplemental funding for the follow up work requiring DNA extraction kits, sequencing, and associated labor. During this time APC counts decreased by a mean of about 1 log(CFU/g) for swabs and about 1.5 log(CFU/g) for produce leaf samples (Appendix A). Aliquots from store samples were stored in 15 mL centrifuge tubes at −20 °C for approximately one and a half months prior to DNA extraction. However, of the store samples, only the produce leaves were later sequenced since the swabs did not have high enough DNA concentrations to likely obtain successful sequencing (i.e., below the 10 ng/µL suggested by the DNA sequencing center).

### 2.5. Aerobic Plate Counts, Coliform Counts, and E. coli

Aerobic plate counts (APCs) and coliform counts were done as an indicator of quality. Generic *E. coli* was tested as a food safety indicator. To obtain aerobic plate counts, coliform counts, and *E. coli* presence from field samples prior to cold storage, PBS wash serial dilutions were plated onto respective media immediately following processing. Standard methods agar (SMA) (Hardy Diagnostics, Santa Maria, CA, USA) was used for APCs, violet red bile agar (VRBA) (GranuCult, MilliporeSigma, Burlington, MA, USA) was used for coliform counts, and 3M^TM^ Petrifilm EC (3M, Saint Paul, MN, USA) was used for *E. coli* (Appendix A). For the SMA and VRBA, 100 µL of tenfold serial dilutions were plated. SMA plates were incubated at 35 °C for 48 ± 2 h. All colony counts reported are log10 transformed. The upper limit of quantification was 9.40 log(CFU/g). VRBA plates were incubated at 35 °C for 18 to 24 h. The limit of detection was 2.00 log(CFU/g). For the Petrifilms, 1 mL of the tenfold serial dilutions were plated. Petrifilms were incubated at 35 °C for 48 ± 4 h. The limit of detection was 1.00 log(CFU/g).

### 2.6. DNA Extraction

DNA was extracted for microbial community profiling. While APCs and coliform counts are useful for determining the quantity of bacteria present, they do not provide information about the specific types of bacteria present. The data just on the quantity of aerobic bacteria and coliforms may not be sufficient to make an informed conclusion on the similarity between swab and produce leaf samples. Therefore, microbial community profiling was done to further investigate the types of bacteria in each sample type and to ensure that the swabs were able to recover a variety of bacteria representative of the community present on the produce leaves. Microbial community profiling was performed as a retrospective analysis of opportunity. DNA extraction was done from the PBS aliquots that were saved when samples were initially processed and were then stored refrigerated for approximately three months. The authors acknowledge that the cold storage prior to extraction is a weakness as it likely affected the relative abundance of some organisms, namely *Pseudomonas*. However, this was an opportunistic analysis used to justify further research on aggregative swab sampling of leafy greens. The authors believe these data have some value as there is no obvious reason to think that the three-month storage would differently affect sample types. Therefore, the data would still be relevant to the primary question of the aggregative swabs’ performance compared to composite produce leaf samples.

For the DNA extraction, the PBS aliquots were used to obtain a pellet of cells. To pellet the cells, 5 mL of the aliquot were centrifuged in 15 mL falcon tubes at room temperature for 15 min at 4000 rpm. The supernatant was decanted, and the remaining pellet was resuspended with 800 µL of C1 solution from the Qiagen DNeasy PowerSoil Pro kit (QIAGEN, Hilden, Germany). The resuspended pellet was added to the PowerBead tubes from the kit. Instructions from the January 2020 DNeasy PowerSoil Pro Kit Handbook were followed with one modification to the bead beating step. Briefly, the modification entailed placing the PowerBead tubes in the Mini-BeadBeater 96 (BioSpec Products, Inc., Bartlesville, OK, USA) and running for 60 s followed by a rest on ice for 1–2 min. This was repeated 4 more times for a total of 5 min of bead beating. The DNA was then stored at −20 °C until sequencing. Samples bought from the store followed the same procedure but used a pellet obtained from 10 mL of sample aliquot to account for lower plate counts and therefore a lower number of input cells. 

### 2.7. DNA Quality Assessment

To obtain 260/280 and 260/230 ratios to indicate if extractions were “pure” DNA and if there was presence of organic compounds, respectively, 1 ul of DNA extract was analyzed on the NanoDrop 2000c Spectrophotometer (Thermo Scientific^Tm^, Waltham, MA, USA). To check DNA base pair length, gel electrophoresis was done. The gel was a 1% agarose gel containing SYBR Safe gel stain (Invitrogen^TM^, Waltham, MA, USA) and was run at 90V for 1 h. A 2-log ladder ranging from 0.1 to 10 kb (New England Biolabs, Ipswich, MA, USA) was run alongside the samples to determine DNA length. Gels were visualized using blue light on the Bio-Rad Gel Doc^TM^ EZ Imager (Bio-Rad Laboratories, Inc., Hercules, CA, USA). DNA quality was found to be appropriate for sequencing. 

### 2.8. DNA Quantification

Quant-iT PicoGreen dsDNA reagent (Invitrogen, Waltham, MA, USA) was used to obtain DNA concentrations. The DNA exacts were diluted either 1:50 or 1:100 with 1X TE depending on the concentration given by NanoDrop to ensure that the concentration reading would fall within the standard curve. For the standard curve, five different concentrations were made by diluting 100 µg/mL stock lambda DNA standard with 1X TE. The concentrations were 1000 ng/mL, 500 ng/mL, 100 ng/mL, 10 ng/mL, and 1 ng/mL. A blank of 1X TE was also included to create a six-point standard curve. Each time samples were run, and the standards were run twice: one set in the first column of the 96 well black clear bottom plate and one in the last column of the plate. For each standard and sample, 100 µL of the standard or sample and 100 µL of PicoGreen reagent were added to the wells and mixed. The plate was allowed to incubate at room temperature protected from light for 2–5 min. The plate was read on the FilterMax F5 Multi-Mode Microplate Reader (Molecular Devices, LLC, San Jose, CA, USA) with wavelengths of 485 nm and 535 nm. A log-log standard curve was created using the log concentration in ng/mL and log fluorescent values. Concentrations of the samples were determined from the standard curve, and the units were converted to ng/µL, as this was the unit used by the sequencing center to report the suggested DNA concentration range. Sample concentrations ranged from 8.00 ng/µL to 131.56 ng/µL.

### 2.9. DNA Submission

Samples above 50 ng/µL were diluted with C6 solution to obtain concentrations within the range of 10–50 ng/µL requested by the sequencing center. A 15 µL volume of each sample was submitted along with two negative controls of 15 µL of PCR water and 15 µL of elution from a DNA extraction performed with no input sample or DNA included. Samples were brought to the Roy J. Carver Biotechnology Center DNA Services Laboratory at University of Illinois at Urbana-Champaign for library preparation and full length 16S rRNA sequencing using PacBio Sequel IIe (Pacific Biosciences, Menlo Park, CA, USA).

### 2.10. Library Preparation and Sequencing

The 16S amplicons were barcoded and generated with full-length 16S primers from PacBio (forward sequence: AGRGTTYGATYMTGGCTCAG, reverse sequence: RGYTACCTTGTTACGACTT) and the 2× Roche KAPA HiFi Hot Start Ready Mix (Roche, Basel, Switzerland). The amplicons were converted to a library with the SMRTBell Express Template Prep kit 2.0 (Pacific Biosciences, Menlo Park, CA, USA). The library was sequenced on one SMRTcell 8M on the PacBio Sequel IIe using CCS sequencing mode and 10-h movie time. 

### 2.11. Filtering, Trimming, and Obtaining Amplicon Sequence Variants (ASVs)

CCS analysis was done using SMRTLink V10.0 with parameters of 3 CCS minimum passes and minimum read quality of 0.999. A total of 965,826 reads were obtained. Reads were demultiplexed into fastq files using lima. Processing of demultiplexed PacBio amplicon sequencing data into chimera-free amplicon sequencing variants (ASVs) was done essentially as in Callahan, et al. (2019) [24]. Briefly, the DADA2 package in R [25] was used to remove primers, filter out reads with less than 1200 bp or greater than 1600 bp as used in Johnson, et al. (2019) [26], and to learn and remove sequencing errors. This resulted in 857,485 reads from which 3422 ASVs were obtained. Chimeras were then removed, so the final output of the pipeline was 3298 non-chimeric ASVs.

### 2.12. Assigning Taxonomy

The 16S Silva version 138.1 with species DADA2-formatted reference database updated 10 March 2021 [27] was used to assign taxonomy. The database is known to have a problem in 3/895 families and 59/3936 genera. Assigning taxonomy was done in R (R version 4.1.2) using the DADA2 package (version 1.20.0) which implements the RDP Naïve Bayesian Classifier algorithm with kmer size 8 and 100 bootstrap replicates. 

### 2.13. Rarefication of Reads

Samples had a range of 9,179 to 42,936 reads with an average of 31,156 reads. Samples were rarefied before determining alpha and beta diversity, investigating the top taxa present, and performing differential abundance analysis. Each sample was rarefied to 7322 reads. During the rarefication step, 203 ASVs were removed since they were no longer present in any sample after random subsampling. 

### 2.14. Determining Alpha and Beta Diversity

Alpha and beta diversity were both determined using the phyloseq R package (version 1.36.0) [28]. Shannon index was used as the measurement for alpha diversity and Bray–Curtis distance and principal coordinates analysis was used for beta diversity.

### 2.15. Obtaining Top Taxa

To obtain the top taxa present in swabs and produce leaf samples, swabs and produce leaves were subset into separate lists, taxa of the same name were agglomerated, and the top five most abundant taxa for each taxonomical rank were obtained. The remaining taxa that were not a part of the top five were grouped together to form the “other” category.

### 2.16. Differential Abundance Analysis

A differential abundance analysis was done using an ANOVA-like differential expression (ALDEx) analysis using Kruskal-Wallis tests with Benjamini-Hochberg corrections and a *p*-value cutoff of 0.05 [29]. The family taxonomical rank was chosen for analysis since it was a lower taxonomical ranking which showed more variation among field produce leaf samples and swabs compared to higher taxonomical ranks. This allowed for the investigation of more differences in microbes among the sample types. 

### 2.17. Statistical Tests

All statistical tests were conducted in R (R version 4.1.2). Specifics on the statistical tests performed are described in the results. 

## 3. Results

### 3.1. Results Overview

This pilot study collected leaf and swab samples from commercial romaine fields and subjected them to indicators of microbial quality (APC, total coliforms) and safety (generic *E. coli*), as well as microbial community analysis (full length 16S). Based on resources available for this pilot study, APCs, coliforms, and generic *E. coli* testing were performed promptly (with 48 h from sample collection), and when additional resources then became available, retrospective testing for microbial community analysis was performed after ~3 month of cold storage. Overall, this pilot study and retrospective analysis show similarity between the leaf tissue grab and swab sampling methods in recovering quality and safety indicators from commercial romaine. These results justify further research with a larger, more rigorous, and more generalizable design.

### 3.2. Swabs and Produce Leaves Recovered Similar Aerobic Plate Counts from Field Romaine

The upper limit of quantification for APCs were 9.40 log(CFU/g). All samples at or above the limit of quantification were analyzed as 9.40 log(CFU/g). The following results are from samples collected from both inoculated and uninoculated sections of the field. Following sample collection, the aerobic bacterial plate counts of swabs (*n* = 12) ranged from 8.28 to 9.40 log(CFU/g) with a mean and standard deviation of 9.21 ± 0.42 log(CFU/g) and median of 9.40 log(CFU/g) (Figure 1). For the swabs, CFU/g refers to CFU per gram of matter picked up by the swab. The APCs of produce leaves taken from the field (*n* = 14) ranged from 8.14 to 9.40 log(CFU/g) with a mean and standard deviation of 9.17 ± 0.43 log(CFU/g) and median of 9.36 log(CFU/g) (Figure 1). Swab sample and produce leaf populations did not have a significant difference in variance as indicated by a F test (*p* = 0.916). A two-way ANOVA of the means showed that sample type (*p* = 0.784), inoculation status (*p* = 0.232), and the interaction term (*p* = 0.381) were not significant. The medians were also not significantly different (*p* = 0.695) by a Mood’s median test. Additionally, swabs and produce leaf outliers in the boxplots were not from the same sections of the field. 

### 3.3. Swabs and Produce Leaves Recovered Similar Coliform Counts from Field Romaine

The following results are from samples collected from both inoculated and uninoculated sections of the field. The coliform counts of swabs ranged from 2.30 to 6.07 log(CFU/g) with a mean and standard deviation of 4.19 ± 1.15 log(CFU/g) and median of 4.72 log(CFU/g) (Figure 2). The produce leaf coliforms ranged from 2.00 to 4.83 log(CFU/g) with a mean and standard deviation of 3.80 ± 0.76 log(CFU/g) (Figure 2). The produce leaf coliform counts had a lower median than swabs at 3.89 log(CFU/g). However, medians were not significantly different (*p* = 0.238) by Mood’s median test. A two-way ANOVA of the means showed that sample type (*p* = 0.303), inoculation status (*p* = 0.837), and the interaction term (*p* = 0.114) were not significant. Swab sample and produce leaf populations did not have significantly different variances as indicated by an F test (*p* = 0.164). 

### 3.4. Swabs Detected E. coli More Often Than Produce Leaf Samples

Of all the samples taken from the field, 8 of 12 swabs detected presence of *E. coli*, while only 3 of 14 composite produce leaf samples detected presence of *E. coli* (Table 1). A Fisher’s Exact test revealed that there was a significant association between *E. coli* presence and sample type (i.e., swabs or produce leaves) (*p* = 0.045). When inoculation status was taken into account, 4 of 6 swabs of romaine from inoculated areas tested positive for *E. coli* and 4 of 6 swabs taken from uninoculated areas tested positive. The *E. coli* concentrations of positive swabs taken from inoculated areas ranged from 1.30 to 5.61 log(CFU/g) while the concentrations of positive swabs from uninoculated areas ranged from 1.00 to 4.38 log(CFU/g). Meanwhile, 3 of 8 produce leaf samples taken from inoculated areas tested positive for *E. coli* and 0 of 6 produce leaf samples from uninoculated areas tested positive (Table 1). The *E. coli* concentrations of positive produce leaf samples taken from inoculated areas ranged from 3.20 to 4.38 log(CFU/g). A Fisher’s Exact test revealed that there was a significant association (*p* = 0.05) between *E. coli* presence and sample type (in this instance, swab sample from inoculated area, swab sample from uninoculated area, produce leaves from inoculated area or produce leaves from uninoculated). Interestingly, the 4 swabs that did not detect *E. coli* were collected 5-days post inoculation, indicating potential die-off. Meanwhile all swabs collected 2- and 0-days post inoculation successfully detected *E. coli*. Of the three produce leaf samples that detected *E. coli*, one was collected two-days post inoculation while the other two samples were collected 0-days post inoculation. With regard to the potential of die-off, a study conducted on romaine grown in New York State using the same three rifampicin-resistant *E. coli* strains used in this study reported an average die-off rate of 0.52 log MPN per plant per day with a 95% confidence interval of 0.17 to 0.87 log MPN [30]. However, it is also important to note that the *E. coli* detected in this study was generic *E. coli*, which may have also come from other sources of contamination, in addition to the sprayed inoculum. 

### 3.5. Swabs and Field Produce Leaves Had Similar Alpha Diversities

The Shannon index used to measure alpha diversity was 4.24 ± 0.28 for swabs and 4.29 ± 0.47 for produce leaves (Figure 3). Meanwhile, the Shannon index for the store produce leaves that was used as an outgroup for the study was 2.49 ± 0.56 (Figure 3). A two-way ANOVA showed sample type and inoculation were significant effects on alpha diversity (*p* < 0.05). The interaction effect approached significance (*p* = 0.058). Upon further investigation, inoculation was not significant by a post hoc Tukey HSD test (*p* = 0.284). A post hoc Tukey HSD test did however show that store produce leaves were significantly different from both swab sample and field produce leaf sample types (*p* < 0.05). Meanwhile, swab sample and field produce leaf alpha diversities were not significantly different from each other (*p* = 0.945). 

### 3.6. Swabs and Produce Leaves Had Different Beta Diversities

Beta diversity was measured using Bray–Curtis distance and principal coordinate analysis (PCoA). The PCoA plot showed that the three sample types formed separate clusters (Figure 4). A two-way permutational multivariate analysis of variation (PERMANOVA) with 999 permutations was done on the Bray–Curtis distance matrix with sample type and inoculation effects. Sample type was significant (*p* = 0.001). Inoculation status and the interaction term were not significant (*p* > 0.05). Pairwise PERMANOVAs with 999 permutations were done as a post hoc analysis. All three sample types had significantly different cluster centroids from each other (*p* < 0.05). 

### 3.7. Swabs and Produce Leaves Had Some of the Most Abundant Taxa in Common

The taxa between sample types were compared to confirm the swabs could pick up and recover a variety of bacteria representative of the community present on the produce leaves. The five most abundant taxa at each taxonomic rank were investigated in the most depth. Some taxa abundances in this study may have been affected by cold storage prior to DNA extraction. The most abundant classes and orders shown in Figure 5b,c are discussed in the Appendix A. For the phylum rank, both swab sample and field produce leaf samples were dominated by the *Proteobacteria* phylum (91.0% and 67.0% on average, respectively). Other phyla that made up the swab sample and produce leaf microbial communities included *Bacteroidota* (4.1% and 14.4%, respectively), *Actinobacteriota* (0.35% and 9.8%, respectively), *Firmicutes* (0.9% and 6.6%, respectively), and *Verrucomicrobiota* (3.6% and 1.8%, respectively). The store produce leaves were also vastly made up of *Proteobacteria* (85.2%), but also included a phylum not present in either field produce leaf or store samples (*Cyanobacteria* (11.0%)). To a lesser extent, the rest of the microbial community of store produce leaves consisted of *Actinobacteriota* (0.2%), *Bacteroidota* (0.1%), and *Firmicutes* (3.6%) (Figure 5a).

The most abundant families shared by swab sample and field produce leaf samples were *Pseudomonadaceae* (30.9% and 24.3% on average, respectively) and *Rhizobiaceae* (6.3% and 5.4%, respectively). Many families in swabs and produce leaves fell into the “other” category (37.7% and 55.7%, respectively). Most abundant families that differed between sample types were *Alcaligenaceae* (5.6%), *Alteromonadaceae* (10.8%), and *Caulobacteraceae* (8.7%) in swabs and *Flavobacteriaceae* (4.6%), *Microbacteriaceae* (4.9%), and *Sphingobacteriaceae* (5.2%) in field produce leaves. Similar to when looking at the orders, swabs had more families in the top five families that belong to the *Gammaproteobacteria* class including *Alteromonadaceae* (10.8%) and *Pseudomonadaceae* (30.9%), while produce leaf samples only had *Pseudomonadaceae* (24.3%). Swabs also had more families that belong to the *Alphaproteobacteria* class. Swabs contained *Caulobacteraceae* (8.7%) and *Rhizobiaceae* (6.3%), while produce leaves only had *Rhizobiaceae* (5.4%). Field produce leaves also had two families, *Flavobacteriaceae* (4.6%) and *Sphingobacteriaceae* (5.2%), that were part of the *Bacteroidetes* phylum while swabs did not have any families from this phylum in the most abundant. Overall, swabs and field produce leaves shared 50 families, produce leaves had 40 families not in swabs, and swabs had 17 unique families not in produce leaves. The store produce leaf outgroup was less diverse with nearly all the families present belonging to only five families and most bacteria present belonging to the *Pseudomonadaceae* family (80.0%) (Figure 5d).

The most abundant genus shared by swabs and field produce leaf samples was *Pseudomonas* (31.1% and 24.5% on average, respectively). Swabs and produce leaves had a diverse array of genera with many genera not in the five most abundant and therefore falling into the “other” category (44.8% and 58.1%, respectively). Most abundant genera that differed between sample types were *Rhizobium* (4.5%), *Brevundimonas* (4.3%), *Phenylobacterium* (4.4%), and *Rheinheimera* (10.9%) in swabs and *Exiguobacterium* (4.2%), *Flavobacterium* (4.5%), *Massilia* (4.2%), and *Pedobacter* (4.4%) in produce leaves. Only looking at the five most abundant genera of each sample type, swabs had more genera belonging to the *Gammaproteobacteria* class with both *Pseudomonas* (31.1%) and *Rheinheimera* (10.9%) compared to produce leaves with *Pseudomonas* (24.5%). Swabs also had genera belonging to the *Alphaproteobacteria* class in the five most abundant (*Rhizobium* (4.5%), *Brevundimonas* (4.3%), and *Phenylobacterium* (4.4%)) while produce leaves did not. Field produce leaves had genera belonging to the *Bacilli* class (*Exiguobacterium* (4.2%)), *Betaproteobacteria* class (*Massilia* (4.2%)), and the *Bacteriodetes* phylum (*Flavobacterium* (4.5%) and *Pedobacter* (4.4%)) in the five most abundant genera while swabs did not. Overall, swabs and field produce leaves shared 107 genera, produce leaf samples had 111 genera not in swabs, and swabs had 35 unique genera not in produce leaves. The swabs and field produce leaves appeared to be more similar to each other than to the store produce leaves. The store produce leaf group was less diverse with many of the genera present belonging to only five genera (“other” category was 1.8%) and most bacteria present belonged to the *Pseudomonas* genus (80.0%) (Figure 5e). The store produce leaves having less variety of taxa present is consistent with their significantly lower alpha diversity compared to the alpha diversities from the swabs and produce leaves from the field.

### 3.8. Enterobacteriaceae Family Was Not Prominent in Swabs and Produce Leaves Regardless of Inoculation

A closer look at the abundance of the *Enterobacteriaceae* family in swabs was performed because the family includes the coliform group of bacteria, which was used as an indicator of sample quality, and *E. coli* which the plants were inoculated with as an indicator of food safety. The nonrarefied read data revealed that the *Enterobacteriaceae* family was in low abundance in both swabs and produce leaves from the field (Appendix A). This family made up on average 0.292% of the microbial community from swabs, 0.124% in field produce leaves, and 0.118% in store produce leaves. Within the *Enterobacteriaceae* family, swabs contained three different genera: *Citrobacter* (0.004%), *Kluyvera* (0.146%), and *Lelliottia* (0.142%). Field produce leaves had two genera within the *Enterobacteriaceae* family: *Kluyvera* (0.120%) and *Lelliottia* (0.004%). In regard to the effect of inoculation, uninoculated produce leaves had a higher percentage of *Enterobacteriaceae* family than inoculated produce leaves (0.255% compared to 0.019%, respectively). The opposite was true for swabs, with swabs taken from inoculated areas having higher percentages of *Enterobacteriaceae* than swabs taken from uninoculated areas (0.460% compared to 0.081%, respectively). 

Microbial community profiling showed that none of the samples from the field contained the genera *Escherichia* or *Salmonella*, which are genera that include potentially harmful pathogens. The only samples that detected *Escherichia* were the three inoculated store produce leaves. In these samples, *Escherichia* made up on average 0.213% of the microbial community. Otherwise, store produce leaf samples, regardless of inoculation status, did not contain any other genera from the *Enterobacteriaceae* family. 

### 3.9. Differential Abundance Analysis Showed Some Families Are Differently Expressed among Sample Types

An ANOVA-like differential expression (ALDEx) analysis indicated 13 differentially expressed families among the 3 sample types (Figure 6). The differentially abundant family found in the highest abundance, *Pseudomonadaceae*, was more highly expressed in store produce leaves. Meanwhile, *Pseudomonadaceae* was still highly abundant in the swabs and field produce leaf samples. The differentially expressed families in swabs from most to least abundant included *Alteromonadaceae*, *Caulobacteraceae*, *Alcaligenaceae*, and *Kaistiaceae*. These families were still abundant in field produce leaves and rarely abundant or undetected in store produce leaf samples. The differentially expressed families in field produce leaf samples from most to least abundant included *Microbacteriaceae*, *Weeksellacea, Micrococcaceae*, *Sphingomonadaceae*, *Devosiaceae*, *Cyclobacteriaceae*, *Sanguibacteraceae*, and *Microtrichaceae*. These families were in low abundance or undetected in swabs and store produce leaf samples, and even in the field produce leaf samples, they each represented 5% or less of the microbial community.

## 4. Discussion

This pilot study showed promising results from multiple measurements, including both quality and safety indicators on romaine lettuce, that support the further development of the aggregative swab sampling method for food safety testing in produce.

### 4.1. Microbial Enumeration and Community Analysis Suggest Aggregative Swabs Are No Less Representative than Produce Leaf Samples

The not significantly different means from APCs, coliform counts, and alpha diversities reinforce that aggregative swabs perform at least not worse than produce leaf sampling. The higher recovery of *E. coli* in swabs show that swabs could even potentially increase the likelihood of detection of some indicators of food safety and pathogens. The not significantly different bacterial recovery counts in this study are consistent with the results obtained from studies of the aggregative sampling methods used on beef trim. Multiple experiments were conducted in beef, and overall, the aggregative swab sampling methods had similar organism recovery compared to the beef trims and were at least as accurate at detecting pathogens or their surrogates [19]. In a follow-up study testing more swab sampling scenarios, there were 32 assays for various pathogen index targets for STEC and *Salmonella*, and there were no occurrences where the N60-based method for sampling beef trims had significantly better recovery [20]. The validation of swabs in beef trim along with the promising results in this pilot study support further studies to validate this method for use in leafy greens. 

### 4.2. Plate Counts Are High but Comparable to Some Other Studies

The APCs found in this study are higher than those presented in some other studies [31,32,33]; however, the values are not unheard of. For example, one study reports the range of average aerobic bacteria counts amongst different leafy green varieties, including lettuce, to be 7.10 to 9.40 log(CFU/g) [34], while another study ranged from 8.30 to 9.20 log(CFU/g) [35]. Furthermore, while another study reported the mean APC of lettuce to be 7.76 log(CFU/g), the maximum count detected was 15.28 log(CFU/g) [36]. It has been previous found that irrigation and rainfall could enhance the risk of manure and soil particles splashing onto the lettuce and therefore impacting the bacteria on the plants [37,38]. The high APCs could also have an impact on product quality and shelf life. In this study, a recent irrigation event before sample collection and the noticeable presence of soil on the samples may explain the relatively higher APCs.

### 4.3. Most Abundant Taxa from Swabs and Produce Leaves Were Similar

Swabs and produce leaves recovered communities that were not significantly different in alpha diversity; however, one measurement that was significantly different between swabs and produce leaf samples was beta diversity. This indicates that the sample types recovered some different types of bacteria. This was investigated further when examining the most abundant taxa that comprised the bacterial community of each sample. However, some of the taxa abundances found in this study may have been affected by cold storage prior to DNA extraction. The presence of *Actinobacteriota, Bacteroidota, Firmicutes,* and *Proteobacteria* as the dominant phyla in both swabs and field produce leaf samples are consistent with several other studies on the microbial communities of leafy greens [31,32,33,39,40,41,42,43]. The *Actinobacteria* class was detected from field produce leaf samples and was not amongst the most abundant classes in swabs. This class is often found in soil [44] and may indicate that the produce leaf samples contained a higher amount of soil compared to the swabs. At lower taxonomical ranks, such as the family level, there were some taxa in the five most abundant taxa that differed between swabs and field produce leaves. However, while these bacteria were classified differently, they are commonly found from the same sources. For example, the families of *Alcaligenaceae*, *Alteromonadaceae*, and *Caulobacteraceae* that were in the top most abundant families in swabs and not produce leaves, and *Flavobacteriaceae*, *Microbacteriaceae*, and *Sphingobacteriaceae* that were in the top most abundant families in produce leaves and not swabs, are all families that are commonly found in soil and/or water environments [45,46,47,48,49]. The store produce leaves recovered a less diverse community with a significantly lower alpha diversity and with no family besides *Pseudomonadaceae* making up more than 4.0% of the community. This is potentially due to product entrance into the processing facility or leaf age which have previously been shown to result in lower Shannon indices [32]. 

Additionally, some swabs and field produce leaf samples detected the genus *Massilia*, and this was one of the most abundant genera in field produce leaf samples. *Massilia* has been previously isolated from soil and the rhizosphere of some plants, further supporting that the produce leaf samples contained a higher amount of soil compared to swabs. However, interestingly, it has also been identified as a major component of agricultural aerosols in central California [50], which could suggest a relationship between agricultural products and the surrounding air they grow in [31]. Overall, based on the most abundant taxa, field produce leaf samples appear to detect a higher abundance of bacteria specifically from soil and aerosols. However, swabs were still able to detect these bacteria at lower abundances in addition to other types of bacteria present in soil and water in higher abundances compared to produce leaves. 

### 4.4. Pseudomonas Genus Was Highly Abundant

Furthermore, consistent with other studies on leafy greens is the high abundance of the *Pseudomonas* genus in all sample types [31,32,41,42]. In this study, *Pseudomonas* was found in every sample and was overall the most abundant genus in all sample types. Some species in the *Pseudomonas* genus can be plant pathogens, while other species can cause food spoilage and decreased shelf-life [42,51,52]. It has been found that the *Proteobacteria* phylum [32], and more specifically the *Pseudomonas* genus, can increase after produce washing [42]. This may be one reason why the store produce leaves, which were likely washed before selling, may have had a higher abundance of *Pseudomonas* compared to the samples taken directly from the field with no washing. 

Additionally, the high abundance of *Pseudomonas* among all samples may be partially due to the admittedly long cold storage (3 months at 4 °C) that occurred prior to DNA extraction due to logistical constraints of the pilot. Future work would include immediate DNA extraction or storage at −80 °C. The cold storage may have had an effect on *Pseudomonas* abundances because *Pseudomonas* are psychrotrophic and have been shown to be one of the genera of bacteria to dominate the leafy green microbial communities after cold storage [43]. *Pseudomonas koreensis* was the most abundant *Pseudomonas* species in store produce leaf and field produce leaf and the second most abundant *Pseudomonas* species from swabs. This species was originally isolated from Korean agricultural soil [53]. Species of *Pseudomonas* previously reported to cause spoilage in vegetables [54,55,56,57] were found in both swabs and produce leaves with *P. putida* and *P. fluorescens* being detected in all sample types, *P. marginalis* being found in swabs and store produce leaf, and *P. viridiflava* in produce leaves from the field. Both swabs and field produce leaves also detected the species *P. syringae*, which has been reported as a plant pathogen [52]. Biotic lesions caused by plant pathogens such as some of the *Pseudomonas* species [58] or by leaf damage during harvest [42] can act as a potential route for internalization of human pathogens like *E. coli* O157:H7 into plant leaves [58,59]. However, while some samples detected generic *E. coli* using Petrifilms, no *E. coli* were detected from the sequencing data in this study. Overall, the swabs were able to successfully recover a variety of *Pseudomonas* species involved in both spoilage and leaf damage from the surface of the romaine plants which were also recovered from the plant leaves itself.

### 4.5. Some Beneficial Bacteria Were Present in Samples

In addition to spoilage organisms, leafy greens may also contain lactic acid bacteria as part of the microbial community. However, in this study the lactic acid bacteria genera of *Aerococcus, Enterococcus, Lactobacillus, Lactococcus, Leuconostoc, Streptococcus*, and *Weissella* were not detected in any samples including the store produce leaf samples. In another study of leafy green surface microbial communities, these genera were only present in very low abundances [41]. However, they were investigated here because lactic acid bacteria and their metabolites may have some positive effects on fresh vegetables through potentially inhibiting growth of human foodborne pathogens [60,61]. While lactic acid bacteria were not detected, other beneficial bacteria that may compete with or inhibit pathogen growth were detected. For example, bacterial taxa known for their beneficial effects, such as the *Rhizobiales* and *Burkholderiales* orders [62], were present in similar abundances and were two of the most abundant orders in both swabs and field produce leaves. Cardinale et al. (2014) concluded that a high diversity of bacterial taxa and high abundances of beneficial bacteria could intensify the barrier effect against pathogens [62]. These might be two factors that helped contribute to the low abundances of the *Enterobacteriaceae* family found in all samples in this study. Overall, the aggregative swab sampling method recovered a microbial community relatively representative of that from the produce leaves which both contained similar abundances of lactic acid bacteria, spoilage bacteria, and bacteria from the same taxonomical classes.

### 4.6. Some Families Were Differentially Abundant between Sample Types

Four of the eight families differentially abundant in field produce leaf samples, *Microbacteriaceae*, *Micrococcaceae*, *Microtrichaceae*, and *Sanguibacteraceae*, are part of the *Actinobacteriota* phylum. This further supports that field produce leaves may contain a higher amount of soil compared to the swabs. With the exception of *Weeksellaceae* in produce leaves, which belongs to the *Bacteroidota* phylum, the other differentially expressed families from both swabs and field produce leaves belonged to the *Proteobacteria* phylum. Within this phylum, the bacterial families more abundant in swabs belonged to the *Alphaproteobacteria, Betaproteobacteria,* and *Gammaproteobacteria* classes. The bacterial families more abundant in produce leaves belonged to the *Alphaproteobacteria* class. Similar to when investigating the most abundant bacteria, the similarity among swabs and produce leaves deviates more at lower taxonomical ranks and are typically found from environmental sources. 

It is unclear exactly why some taxa are more enriched or unique to a particular sample type. Some hypotheses that could be tested in future studies but are outside the scope of this pilot study, may include differences in the area of the plant being sampled or differences in bacterial attachment on swabs versus on produce leaves. For example, when taking produce leaf samples, one could grab the outer leaves or reach into the top of the plant to take leaves closer to the plant interior. Meanwhile, swabs represent more of the outer sides and tops of the plants. The exterior and interior parts of the plants may contain different bacteria or have varying chances of containing a pathogen. For example, the outer leaves of romaine have previously been shown to be more exposed to environmental conditions and in more direct contact with soil, that can make the plant exteriors more likely to be contaminated [63,64]. On the other hand, bacteria have also been shown to internalize within romaine and other leafy greens [65,66,67,68]. Regarding bacterial attachment, a study on swabbing pork, although with a swab made of a different material than used in this study, explained that the abrasiveness of sampling and the bacteria’s attachment capability to swabs can impact the effectiveness of bacterial removal from pork [69]. Furthermore, Singh et al. (2015) explained that while loosely associated bacteria are picked up when swabbing chicken, it seems like there are some tightly associated bacteria that might only be detected by grinding the tissue [70]. These factors may contribute to some of the variation among swabs and produce leaves at lower taxonomical ranks. 

## 5. Conclusions

In theory, the aggregative swab sampling method could outperform composite produce leaf sampling due to its ability to be more representative of the field compared to produce leaf samples. For example, the swab sample can be passed over a field to create a sample that is representative of hundreds to thousands of plants, collected in a nondestructive manner. In contrast, compositing grabs of produce leaves typically represents only around 60 plants. A sample that is representative of a larger portion of the field could have a better chance of detecting pathogens or other rare bacteria of interest if present within the field. Additionally, the swabs produce a sample with less mass than produce leaf samples, potentially reducing laboratory costs. 

Based on data from this pilot study, swabbing the outer surface of romaine lettuce, the aggregative swabs were at least as effective as produce leaf samples in recovering quality and safety indicators and showed promising results in detecting generic *E. coli*. While the swabs and produce leaves recovered some different bacterial taxa, swabs overall had a community similar to that of produce leaf samples with several shared phyla and classes in similar abundances and with the taxa that differed tending to be from the same environmental sources. Future work should include further method development and validation for the aggregative swabs, including but not limited to, methods to swab more interior romaine leaves that are directly consumed, testing in other commodities like spinach where outer leaves are directly consumed, and validation by direct pathogen testing.

## Figures and Tables

**Figure 1 foods-13-03080-f001:**
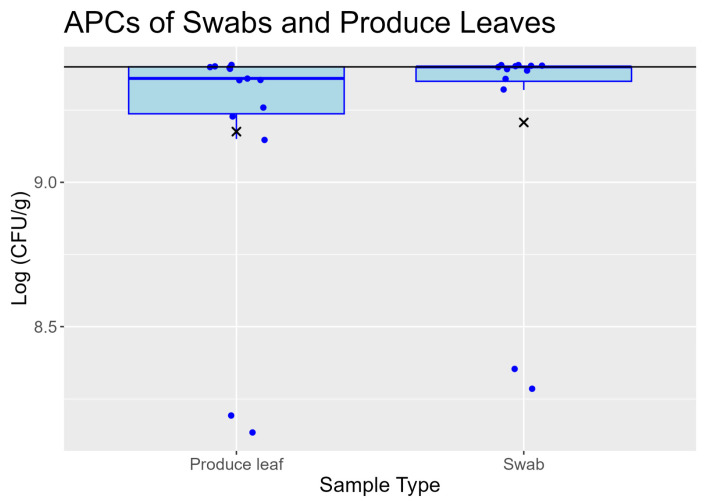
Boxplots comparing swab sample and produce leaf aerobic plate counts from field samples prior to storage. Solid black line represents the upper limit of quantification of 9.40 log(CFU/g). Samples above the limit of quantification were plotted as 9.40 log(CFU/g). For swabs, CFU/g refers to CFU per gram of matter picked up by the swabs. Data points are jittered so all points are visible; therefore, some points may appear slightly above limit of quantification. The middle lines of the boxplots are the medians, and the “×” are the arithmetic means. The lower and upper hinges represent the 25th and 75th percentiles. Whiskers extend to the largest or smallest values that are not beyond 1.5 × QR from the hinge. Data points beyond the whiskers are plotted individually as outliers.

**Figure 2 foods-13-03080-f002:**
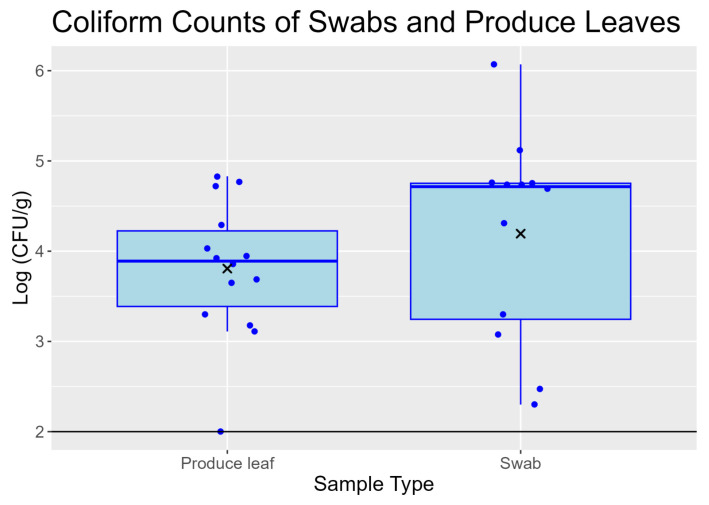
Boxplots comparing swab sample and produce leaf coliform counts from field samples prior to storage. Solid black line represents the lower limit of quantification of 2.00 log(CFU/g). For swabs, CFU/g refers to CFU per gram of matter picked up by the swabs. Data points are jittered so all points are visible. Boxplots are constructed as in Figure 1.

**Figure 3 foods-13-03080-f003:**
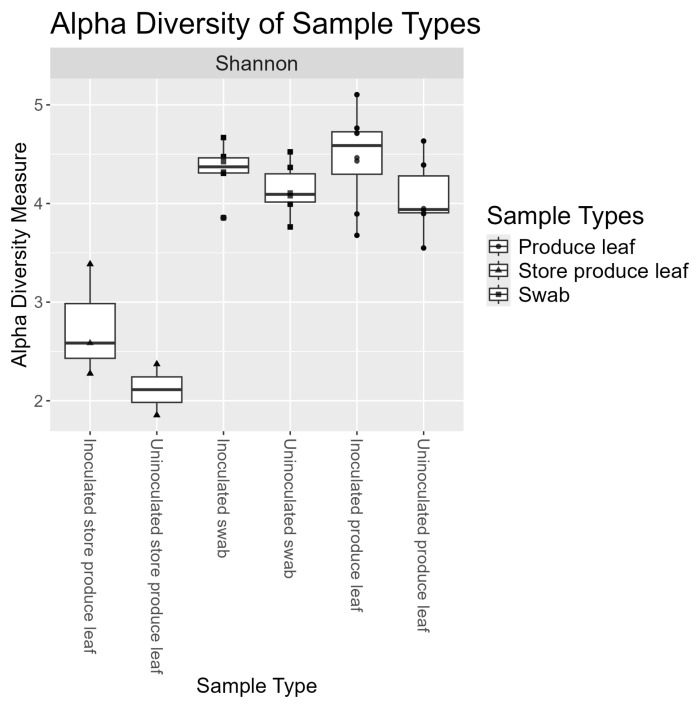
Boxplots of alpha diversity measures using Shannon Index and data rarefied to 3095 ASVs and 226,982 reads. The middle lines of the boxplots are the medians. The lower and upper hinges represent the 25th and 75th percentiles. Whiskers extend to the largest or smallest values that are not beyond 1.5 × QR from the hinge. Data points beyond the whiskers are plotted individually as outliers.

**Figure 4 foods-13-03080-f004:**
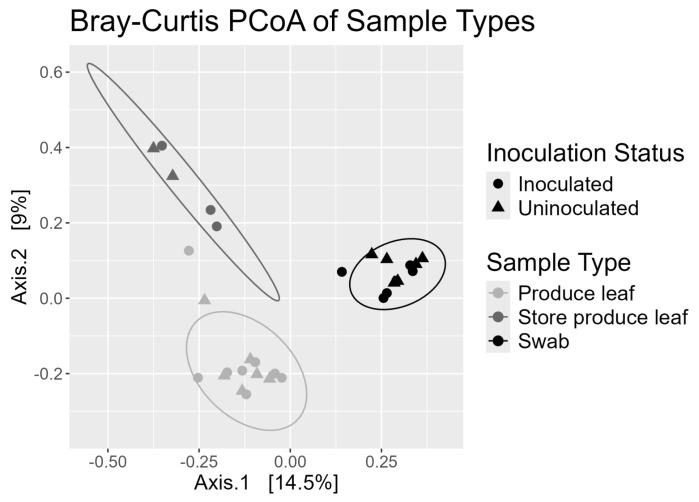
Principal coordinates analysis plot of store produce leaves, swabs, and field produce leaf samples using Bray–Curtis Distance and data rarefied to 3095 ASVs and 226,982 reads. Inoculation status of samples are differentiated by the shape of the data points and sample types are differentiated by color. Ellipses around points represent a 95% confidence ellipse based on a multivariate t-distribution.

**Figure 5 foods-13-03080-f005:**
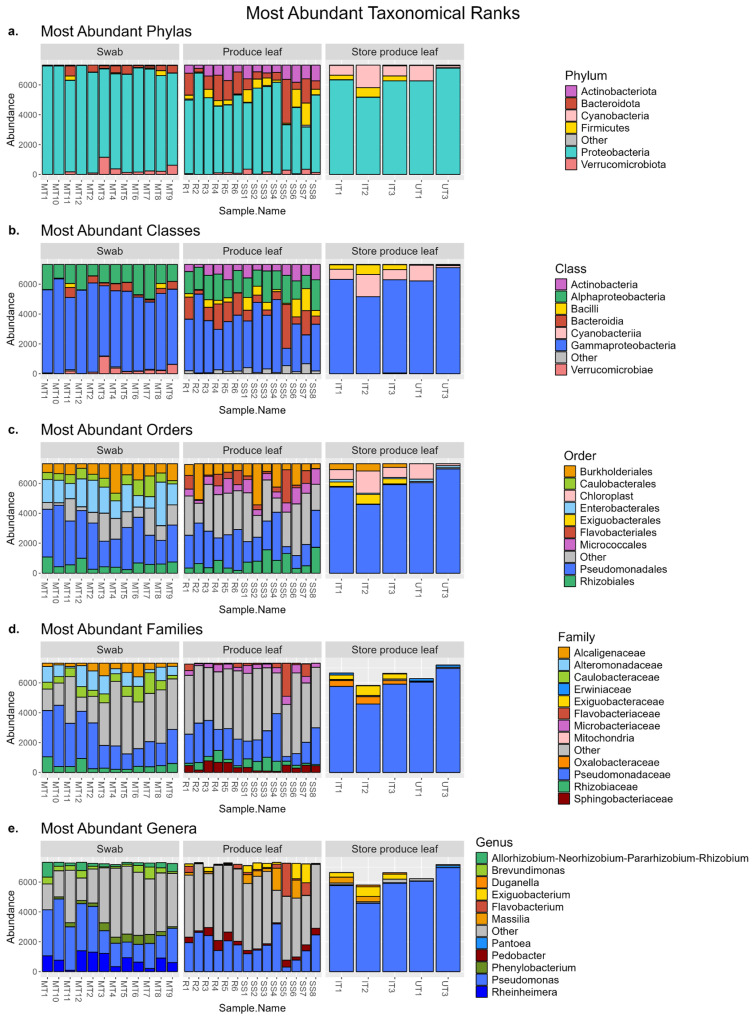
Top 5 most abundant taxa in store produce leaves, swabs, and field produce leaves at the taxonomical ranks of phylum (**a**), class (**b**), order (**c**), family (**d**), and genus (**e**). All taxa not in the 5 most abundant for that sample type are represented in the ‘Other’ category. Taxa that fall within the phylum *Actinobacteriota* are represented by a purple color, *Bacteroidota* are shades of red, *Firmicutes* are yellow, and *Verrucomicrobiota* are pink in (**a**) through (**e**). Taxa that fall within the *Alphaproteobacteria* class are represented by shades of green in (**b**) through (**e**), the *Gammaproteobacteria* class are shades of blue in (**b**) through (**e**) and the *Betaproteobacteria* class are shades of orange in (**c**) through (**e**). The *Proteobacteria* phylum in (**a**) includes classes *Alphaproteobacteria*, *Gammaproteobacteria*, and *Betaproteobacteria* and is represented by a blue/green color.

**Figure 6 foods-13-03080-f006:**
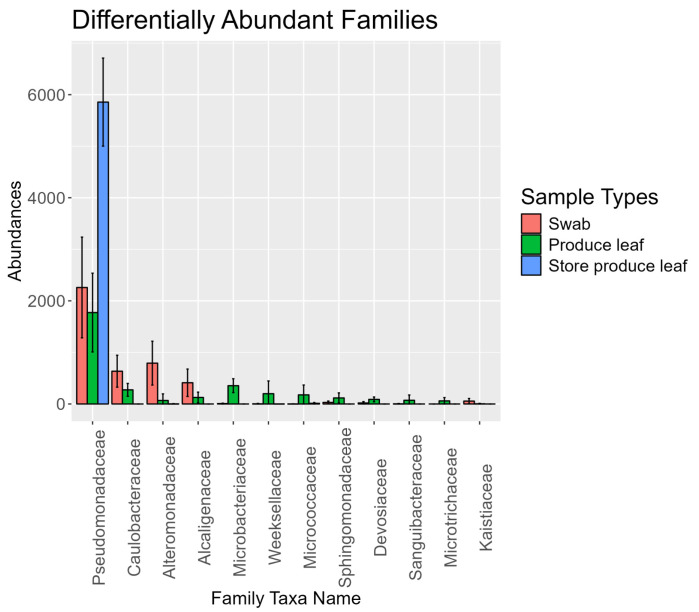
Bar plot of the abundances of each of the differentially abundant families determined using ANOVA-like differential expression (ALDEx) using Kruskal–Wallace tests, Benjamini–Hochberg corrections, and a *p*-value cutoff of 0.05. The bar heights represent the mean abundance and error bars represent a 95% confidence interval. Rarefied data were used. Each sample was rarefied to 7322 reads.

**Table 1 foods-13-03080-t001:** Contingency table showing the number of swabs and produce leaf samples that were positive for generic *E. coli* based on results from 3M^TM^ Petrifilms. The limit of detection was 1 log(CFU/g).

Sample Type	Inoculation Status	*E. coli* Positive	*E. coli* Negative	Total
Swabs	Inoculated	4	2	6
	Uninoculated	4	2	6
Produce leaf	Inoculated	3	5	8
	Uninoculated	0	6	6
Total		11	15	26 *

* Fisher’s Exact test: *p* = 0.05 for overall comparison between swabs from inoculated areas, swabs from uninoculated areas, produce leaves from inoculated areas, and produce leaves from uninoculated areas.

## Data Availability

The data have been deposited to BioProject accession number PRJNA824469 in the NCBI BioProject database, https://www.ncbi.nlm.nih.gov/bioproject/ (accessed 17 September 2024). The code for processing and analysis of DNA sequences can be found on GitHub, https://github.com/rgathm2/Gathman_romaine_swabs.git (accessed 17 September 2024).

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
