# Peer review of "Aggregative Swab Sampling Method for Romaine Lettuce Show Similar Quality and Safety Indicators and Microbial Profiles Compared to Composite Produce Leaf Samples in a Pilot Study"

_foods, 2024, doi:10.3390/foods13193080_

Round 1
Reviewer 1 Report
Comments and Suggestions for Authors
This manuscript discusses the potential advantages of using aggregative swab sampling over composite produce leaf sampling in agricultural settings. The authors measured aerobic plate counts and coliforms in both sampling types from a one-acre field and found no significant difference in mean values or variance between the two methods.
In their experimental plan, authors used both uninoculated and inoculated fieds with a cocktail of E. coir rif+ strains. Globally, aggregative swabs were more effective at detecting generic E. coli. Yet, authors didn’t prove that the inoculation they used can be retrieved while sampling.
Microbial profiling using full-length 16S rRNA sequences showed similar alpha diversities and prevalence of the most common bacterial taxa between the two sampling methods. The raw reads data are shared through ncbi and the R scripts are shared on github.
The author’s conclusions are quite well supported by data and well discussed: Swab sampling could be more representative of the entire field as it can sample across hundreds to thousands of plants nondestructively, compared to produce leaf sampling which ( sixty plants). This broader sampling could enhance the detection of pathogens or other rare bacteria within the field. Swabs also produce samples with less mass, potentially reducing lab costs. Future research is recommended to further develop and validate aggregative swab methods.
Minor comments
Line 32. This percentage refers to outbreaks. So not sure than applying to the total burden of foodborne the percentages observed in outbreaks are enough (e.g. campylobacteriosis's main source is poultry while we don’t see any outbreak with poultry products because mainly sporadic cases are generated)
Line 43 STEC (not E. coli). More generally related to romaine lettuce provide the serotype (O26?) and sub-types characteristics (eae, stx2a?)
Line 72 “if 1% of these individual samples would test positive” not sure about that. It should rather be “if the real prevalence would be 1%, N60 result in %x chance of detecting the batch as positive”, isn’t it?
Line 120 “zero tolerance” is a poor description of the reality (sometimes means “zero risk”). It’s never zero risk, any wording different? A real sampling plan based on risk would use Codex metrics (ALOP, FSO, PO,…) and from PO it may be possible to derive risk-based microbiological criteria.
Lines 157-160. All these are not SI units. Check with mdpi author’s instruction but SI units are maybe advisable
Figure 1. Geometric or arithmetic means? (for ‘x’)
Another point: maybe use log10 to avoid any misunderstanding related to CFU
Author Response
|
Response to Reviewer 1 Comments
|
||
|
1. Summary |
|
|
|
Thank you very much for taking the time to review this manuscript. Please find the detailed responses below and the corresponding revisions/corrections highlighted/in track changes in the re-submitted files.
|
||
|
2. Questions for General Evaluation |
Reviewer’s Evaluation |
Response and Revisions |
|
Does the introduction provide sufficient background and include all relevant references? |
Yes |
|
|
Is the research design appropriate? |
Yes |
|
|
Are the methods adequately described? |
Yes |
|
|
Are the results clearly presented? |
Yes |
|
|
Are the conclusions supported by the results?
|
Yes |
|
|
3. Point-by-point response to Comments and Suggestions for Authors |
|
|
|
Comments 1: Line 32. This percentage refers to outbreaks. So not sure than applying to the total burden of foodborne the percentages observed in outbreaks are enough (e.g. campylobacteriosis's main source is poultry while we don’t see any outbreak with poultry products because mainly sporadic cases are generated)
|
||
|
Response 1: Thank you for pointing this out. We added an estimate on foodborne illnesses attributed to romaine and the economic burden they cause.
|
||
|
Comments 2: Line 43 STEC (not E. coli). More generally related to romaine lettuce provide the serotype (O26?) and sub-types characteristics (eae, stx2a?) |
||
|
Response 2: We added the names of the STEC serotypes that have caused STEC outbreaks in leafy greens and included the percentage of the leafy green STEC outbreaks each serotype caused.
Comments 3: Line 72 “if 1% of these individual samples would test positive” not sure about that. It should rather be “if the real prevalence would be 1%, N60 result in %x chance of detecting the batch as positive”, isn’t it?
Response 3: Sentence was revised to say “if 1% of the samples were contaminated” rather than saying 1% would test positive.
Comments 4: Line 120 “zero tolerance” is a poor description of the reality (sometimes means “zero risk”). It’s never zero risk, any wording different? A real sampling plan based on risk would use Codex metrics (ALOP, FSO, PO,…) and from PO it may be possible to derive risk-based microbiological criteria.
Response 4: The term “zero tolerance” in the context used in the manuscript does not refer to the sampling plan of the study. The term “zero tolerance” as used was meant as the FDA definition meaning that if a product contains the specified substance or organism that is considered potentially dangerous, the produce is considered unfit for human consumption. Clarification on what was meant by “zero tolerance” was added to the text.
Comments 5: Lines 157-160. All these are not SI units. Check with mdpi author’s instruction but SI units are maybe advisable.
Response 5: Thank you for pointing this out. SI units have been added.
Comments 6: Figure 1. Geometric or arithmetic means? (for ‘x’)
Response 6: Added clarification in the figure 1 legend that the plotted means are the arithmetic means.
Comments 7: Another point: maybe use log10 to avoid any misunderstanding related to CFU
Response 7: Added clarification in the methods section that all colony counts that are reported were log10 transformed.
|
||
|
4. Response to Comments on the Quality of English Language |
||
|
Point 1: English language fine. No issues detected. |
||
|
|
||
|
|
||
Reviewer 2 Report
Comments and Suggestions for Authors
L64 reference to "Machado‐Moreira, B., Richards, K., Brennan, F., Abram, F. and Burgess, C.M., 2019. Microbial contamination of fresh produce: what, where, and how? Comprehensive reviews in food science and food safety, 18(6), pp.1727-1750" was mentioned due to low concentrations of contamination. However, this paper only mentions outbreak numbers and, therefore, is not relevent to this statement. As this statement justifies this study's novelty, this needs to be changed to a relevant study. This paper is similar to "Quintanilla Portillo, J., Cheng, X., Belias, A.M., Weller, D.L., Wiedmann, M. and Stasiewicz, M.J., 2022. A validated Preharvest sampling simulation shows that sampling plans with more randomly located samples perform better than typical sampling plans in detecting representative point-source and widespread hazards in leafy Green fields. Applied and environmental microbiology, 88(23), pp.e01015-22." The differences must be highlighted for this paper to be novel. The samples were tested three months after cold storage, a process that was not validated and resulted in many Pseudomonas, which could cause variation in the results. Justification for their presence in this paper is not valid without further investigation.
L199, only one strain was used for an outgroup, whereas a cocktail was used for L161 (set up); these studies cannot be compared. This leaves the data lacking for this study.
L380 fig 1 shows a box plot comparing swab sample and leaves from field trials with only aerobic plate counts, and no E coli or pseudomonas was mentioned. E coli or pseudomonas were only tested after the 3 months were in the discussion but not the results. Table 1 is, therefore, confusing.
Author Response
|
Response to Reviewer 2 Comments
|
||
|
1. Summary |
|
|
|
Thank you very much for taking the time to review this manuscript. Please find the detailed responses below and the corresponding revisions/corrections highlighted/in track changes in the re-submitted files.
|
||
|
2. Questions for General Evaluation |
Reviewer’s Evaluation |
Response and Revisions |
|
Does the introduction provide sufficient background and include all relevant references? |
Must be improved |
Introduction was edited to provide more background on illnesses and burden caused by leafy greens and the novelty of the study. More relevant references were cited. |
|
Is the research design appropriate? |
Must be improved |
Additional acknowledgement and justification of the weakness in design was added to methods section |
|
Are the methods adequately described? |
Must be improved |
|
|
Are the results clearly presented? |
Must be improved |
Additional detail added to results section to help avoid confusion |
|
Are the conclusions supported by the results?
|
Must be improved |
|
|
3. Point-by-point response to Comments and Suggestions for Authors |
|
|
|
Comments 1: L64 reference to "Machado‐Moreira, B., Richards, K., Brennan, F., Abram, F. and Burgess, C.M., 2019. Microbial contamination of fresh produce: what, where, and how? Comprehensive reviews in food science and food safety, 18(6), pp.1727-1750" was mentioned due to low concentrations of contamination. However, this paper only mentions outbreak numbers and, therefore, is not relevent to this statement. As this statement justifies this study's novelty, this needs to be changed to a relevant study. This paper is similar to "Quintanilla Portillo, J., Cheng, X., Belias, A.M., Weller, D.L., Wiedmann, M. and Stasiewicz, M.J., 2022. A validated Preharvest sampling simulation shows that sampling plans with more randomly located samples perform better than typical sampling plans in detecting representative point-source and widespread hazards in leafy Green fields. Applied and environmental microbiology, 88(23), pp.e01015-22." The differences must be highlighted for this paper to be novel.
|
||
|
Response 1: Thank you for pointing this out. The sentence regarding low concentrations of contamination was edited and more appropriate references about sampling plans and microbiological testing and criteria were cited instead.
While this paper and the paper from Quintanilla et al. both contain inoculated field trials where leafy greens are collected in stratified and systematic patterns, this paper is the first report of collecting and using aggregative swab samples for food safety testing in leafy greens. The purpose of the inoculated field trials in Quintanilla et al were to validate a simulation model. No aggregative swabs were collected. The purpose of this paper was to provide proof-of-concept for the use of aggregative swabs for food safety testing in romaine. The difference between the two papers was added to the introduction, and the novelty of this paper was highlighted. Additionally, a paper on a similar aggregative sampling method for soil from romaine fields was cited, and the novelty of this paper being the first reported use of an aggregative swab being collected directly from the leafy green tissue itself was highlighted.
Comments 1 (Second part): The samples were tested three months after cold storage, a process that was not validated and resulted in many Pseudomonas, which could cause variation in the results. Justification for their presence in this paper is not valid without further investigation.
Response 1 (Second Part): We acknowledge that the cold storage prior to DNA extraction is a weakness of the study as it likely affected the relative abundance of some organisms, many Pseudomonas. Microbial community profiling was performed as a retrospective analysis of opportunity. This analysis was used to justify further research on aggregative swab sampling of leafy greens. The authors believe this data has some value as there is no obvious reason to think that the three month storage would differently affect sample types. Therefore, the data would still be relevant to the primary question of the aggregative swabs’ performance compared to composite produce leaf samples. We believe it is better to publish the data, acknowledging the weaknesses of the design, rather than having the data stay within our group where no one else would be able to see it. Further acknowledgement of this weakness in the study design and justification of why the data was included in the paper was added to the methods section.
|
||
|
Comments 2: L199, only one strain was used for an outgroup, whereas a cocktail was used for L161 (set up); these studies cannot be compared. This leaves the data lacking for this study.
|
||
|
Response 2: The results of the outgroup were presented in the results section to highlight the similarity between the swab samples and the produce samples from the field. The comparison of the outgroup results to the swab and the field produce samples were not used to determine the conclusions of the study nor the next steps. Additional detail was added to the text to explain the purpose of the outgroup.
Comments 3: L380 fig 1 shows a box plot comparing swab sample and leaves from field trials with only aerobic plate counts, and no E coli or pseudomonas was mentioned. E coli or pseudomonas were only tested after the 3 months were in the discussion but not the results. Table 1 is, therefore, confusing.
Response 3: E. coli detection and enumeration were performed prior to the cold storage. APCs, coliforms, and generic E. coli testing were performed promptly (with 48 h from sample collection), and when additional resources then became available, retrospective testing for microbial community analysis was performed after ~3 month of cold storage. Broadly this pilot study and retrospective analysis show similarity of leaf tissue grab and swab sampling methods for recovering quality and safety indicators from commercial romaine, results which should then justify follow up study with larger, more rigorous, and more generalizable design. Additional detail was added to the results section for better clarity and to help avoid confusion.
|
||
|
4. Response to Comments on the Quality of English Language |
||
|
Point 1: English language is fine. No issues detected. |
||
|
|
||
|
5. Additional clarifications |
||
|
|
||